# Vibrational and Thermodynamic Properties of Hydrous Iron-Bearing Lowermost Mantle Minerals

**Jiajun Jiang** [1,2]**, Joshua M. R. Muir** [2] **and Feiwu Zhang** [2,*]

1   Faculty of Land Resources Engineering, Kunming University of Science and Technology, Kunming 650093, China; jiangjiajun@kust.edu.cn
2   State Key Laboratory of Ore Deposit Geochemistry, Institute of Geochemistry, Chinese Academy of Sciences, Guiyang 550081, China; j.m.r.muir@mail.gyig.ac.cn
*   Correspondence: zhangfeiwu@vip.gyig.ac.cn

**Abstract:** The vibrational and thermodynamic properties of minerals are key to understanding the phase stability and the thermal structure of the Earth's mantle. In this study, we modeled hydrous iron-bearing bridgmanite (Brg) and post-perovskite (PPv) with different $[Fe^{3+}\text{-H}]$ defect configurations using first-principles calculations combined with quasi-harmonic approximations (QHA). $Fe^{3+}$-H configurations can be vibrationally stable in Brg and PPv; the site occupancy of this defect will strongly affect its thermodynamic properties and particularly its response to pressure. The presence of $Fe^{3+}$-H introduces distinctive high-frequency vibrations to the crystal. The frequency of these peaks is configuration dependence. Of the two defect configurations, $[Fe'_{Si} + OH^-]$ makes large effects on the thermodynamic properties of Brg and PPv, whereas $[V''_{Mg} + Fe^-_{Mg} + OH^-]$ has negligible effects. With an expected lower mantle water concentrations of <1000 wt. ppm the effect of $Fe^{3+}$-H clusters on properties such as heat capacity and thermal expansion is negligible, but the effect on the Grüneisen parameter $\gamma$ can be significant (~1.2%). This may imply that even a small amount of water may affect the anharmonicity of $Fe^{3+}$-bearing $MgSiO_3$ in lower mantle conditions and that when calculating the adiabaticity of the mantle, water concentrations need to be considered.

**Keywords:** hydrogen; bridgmanite; post-perovskite; ferric iron; phonon vibration; thermodynamic property

## 1. Introduction

Iron-bearing bridgmanite (Brg) and its high-pressure phase post-perovskite (PPv) are thought to be the most abundant mineral phases in the Earth's lower mantle and D″ layer along with ferropericlase (Mg, Fe)O and Calcium silicate perovskite $CaSiO_3$ [1,2]. Since the discovery of a phase transition from bridgmanite to post-perovskite [3–5] at relevant pressure and temperature conditions of the lowermost mantle, this has been invoked to explain some seismic features of the D″ layer, such as the origin of D″ and its seismic signals [6–8]. Hence, studying the vibrational and thermodynamic properties of Brg and PPv is critical to revealing the secret of the Brg-PPv phase transition and the thermal structure of the D″ layer. Recent experimental and theoretical studies have shown that the phase transition from Brg to PPv can be very complicated in the presence of defect elements, particularly Fe and Al, as they can both broaden the transition and change the depth at which it occurs [9–11].

While generally, the lower mantle is reductive, Fe can be incorporated into bridgmanite and post-perovskite in both the +2 (ferrous) and +3 (ferric) state, with the latter being favored in the presence of Aluminum at the high pressures and temperatures near the D″ [12]. This is created through a disproportionation reaction of $Fe^{2+}$ into $Fe^{3+}$ and metallic Fe [13–15]. Experimental and theoretical studies demonstrate that $Fe^{3+}$ can be incorporated into the lattice of lower mantle minerals either as $Fe^{3+}$-$Fe^{3+}$ pairs on the Mg (A) and Si (B) sites or as $Fe^{3+}$-$Al^{3+}$ pairs with $Fe^{3+}$ on the A and $Al^{3+}$ on the B though above ~80 GPa these preferences can be changed as some $Fe^{3+}$ swaps to the B site and $Al^{3+}$ to the A site [2,16–19].

Therefore, investigating the effects of impurities (such as $Fe^{3+}$) on the vibrational and thermodynamic properties of bridgmanite and post-perovskite is important for better understanding and modeling the real lower mantle and D″ layer region.

Since the discovery of post-perovskite and the Brg to PPv transition, the vibrational and thermodynamic properties of Mg-bearing bridgmanite and post-perovskite at lower mantle conditions have been explored intensively by experimental [20–22] and theoretical studies [23–26]. Recently, a series of studies have reported a wealth of results demonstrating the effects of Al or Fe on the vibrational and thermodynamic properties of Brg and PPv [27–30]. Those results greatly contribute to our understanding of the thermal structure, composition, and dynamic behavior of the lowermost mantle. However, given the multifarious volatiles present in the lower mantle [31,32], the effects of the complex chemical composition on the thermodynamic properties of the bridgmanite and post-perovskite are still not constrained well, in particular, the effects due to water (H).

Previous studies on vibrational and thermodynamic properties have mainly focused on the dry system. Water (hydrogen), however, is an important volatile in the Earth's mantle and should be considered. It is believed that the transition zone has a high water content with up to ~1–2 wt.%, at least locally [33–35], but water content in the lower mantle is still controversial. Previous studies have suggested that the water content of lower mantle minerals would be no more than ~20 ppm [36,37]; however, some experiments have reported a wider water content range from ~50 ppm to ~0.4 wt.% [38,39]. Recently, experimental and theoretical studies have indicated that the lower mantle minerals, bridgmanite, and post-perovskite, can potentially contain a relatively high amount of water with over 2000 ppm [40,41]. Water in minerals can have significant effects on their physical and chemical properties and thus on those of the Earth's mantle. The presence of water in $MgSiO_3$ will affect the phase stability of bridgmanite and post-perovskite in lower mantle conditions. The phase transition boundary between bridgmanite and post-perovskite shifts to higher or lower pressures, respectively, when hydrogen atoms substitute in the Mg or Si sites in the lattice [42]. Furthermore, water combined with some cations can also influence the seismic properties of Brg and PPv. Our recent study has also found that the elastic properties of the $MgSiO_3$ system, especially the shear velocity, are also remarkably sensitive to the presence of hydrogen in the Fe-bearing Brg and PPv [42]. The incorporation of $Fe^{3+}$ and H in the $MgSiO_3$ system via $[Fe^{3+}\text{-H}]_{Si}$ defect ($Fe'_{Si} + OH^-$) can yield a shear velocity anomaly which is very close to the average anomaly value of LLSVPs from seismic observation. This may imply that hydrous $Fe^{3+}$-bearing $MgSiO_3$ is a dominant mineral in LLSVPs. The effects of water on the physical and chemical properties of $MgSiO_3$, especially for Fe-bearing systems, need to be better constrained. Therefore, the influence of water on the vibrational and thermodynamic properties of bridgmanite and post-perovskite need to be determined further in order to better understand the thermodynamic and thermal structure of the lower mantle.

In this study, we use Density Functional Theory (DFT) in combination with the density functional perturbation theory (DFPT) method to investigate the effects of different hydrogen defect configurations on the vibrational phonon frequency of hydrous $Fe^{3+}$-bearing bridgmanite and post-perovskite. We have also calculated in detail the thermodynamic parameters of hydrous $Fe^{3+}$-bearing lower and D″ layer minerals with different $Fe^{3+}$ and H defect configurations using a quasi-harmonic approximation (QHA), and we have estimated the effects of hydrogen on the thermodynamic properties of $Fe^{3+}$-bearing bridgmanite and post-perovskite.

## 2. Computational Methods

### 2.1. Computational Details

All simulations were carried out with the VASP code [43] using the all-electron projector-augmented-wave (PAW) method [44], and the electron exchange-correlation was described by the PBE form of Generalized Gradient Approximation (GGA) [45]. The used PBE potentials are $1S^2$ core (radius 1.52 a.u.) for O, $1S^2 2S^2$ core (radius 2 a.u.) for

Mg, $1s^2 2s^2 2p^6$ core (radius 1.9 a.u.) for Si, $1s^2 2s^2 2p^6 3s^2 3p^6$ core (radius 2.3 a.u.) for Fe. GGA often cannot correctly describe the electronic structure of iron-containing systems as their d-electrons have strong correlations; therefore, we used a GGA+U functional to treat the iron. The Hubbard U term was set to 3 eV following Hsu et al. [46] and Muir and Brodholt [47], which has already been successfully applied to describe the properties of iron-bearing systems at lower mantle conditions.

For all calculations, we used $(2 \times 2 \times 1)$ and $(4 \times 1 \times 1)$ supercells with 80 atoms for the bridgmanite (Brg) phase and post-perovskite (PPv) phase, respectively. In static runs, the plane-wave cutoff was set to 600 eV, and a $(3 \times 3 \times 4)$ Monkhorst-Pack grid [48] was used for the Brillouin zone sampling. All structures were fully relaxed at an imposed pressure of 30–150 GPa by using the conjugate-gradients method until the total energy differences were less than $10^{-6}$ eV. The force convergence criterion was set to 0.01 eV/Å. After the structural optimizations, frequencies and force constant matrixes were calculated based on the density functional perturbation theory (DFPT) [49]. Phonon dispersion curves were obtained by using the PHONOPY code [50]. Gibbs free energy and high-temperature thermodynamic properties were then calculated using the quasi-harmonic approximation (QHA) method with an $(11 \times 11 \times 11)$ q-point mesh.

## 2.2. The Configurations of Hydrogen Defects and Spin State of Iron

In this study, we considered two schemes of iron-bearing hydrogen defects: $[Fe^{3+}\text{-}H]_{Si}$ defect ($Fe'_{Si} + OH^-$) (0.552 wt.% $H_2O$) and $[Fe^{3+}\text{-}H]_{Mg\text{-}Mg}$ defect ($V''_{Mg} + Fe^-_{Mg} + OH^-$) (0.559 wt.% $H_2O$). These two iron-hydrogen defects configurations correspond to the case where $[Fe^{3+}\text{-}H]$ is in the Si site or the Mg site in the lattice, respectively. Thus, the effects of different occupations of hydrogen and iron on the vibrational and thermodynamic properties of minerals can be examined.

In the iron-bearing system, the spin state of $Fe^{3+}$ needs to be determined to estimate the effect of the $Fe^{3+}$ spin state on the properties of the mineral. For both bridgmanite and post-perovskite, experimental and theoretical studies have reported that the $Fe^{3+}$ at the Mg-site is typically high spin (HS, S = 5/2) throughout the lower mantle region; however, the $Fe^{3+}$ in a Si-site undergoes a spin transition from high spin (HS, S = 5/2) to low spin (LS, S = 1/2) for both bridgmanite [16,18,46] and post-perovskite [17,51]. Therefore, for the $[Fe^{3+}\text{-}H]_{Si}$ defect, we set an initial $Fe^{3+}$ spin state configuration with high spin (HS) and low spin (LS), respectively. For the $[Fe^{3+}\text{-}H]_{Mg\text{-}Mg}$ defect, its initial $Fe^{3+}$ spin state was set to high spin (HS).

HS $Fe^{3+}$ and LS $Fe^{3+}$ can stabilize as a mixed spin state (MS) at lower mantle conditions due to magnetic and configurational entropy at high temperatures. In order to calculate the concentration of high and low spin iron under certain pressures and temperatures, we used the below equation:

$$n_{LS}(P,T) = \frac{1}{1 + \exp\left[\frac{\Delta G_{LS-HS}}{k_B T n_{Fe}}\right]} \quad (1)$$

where $n_{LS}$ is the fraction of the low spin state, $\Delta G$ is the difference between the free energy of low spin and high spin state, $n_{Fe}$ is the number of iron atoms per unit cell and $k_B$ is the Boltzmann constant.

Magnetic entropy was included in the total free energy and was given by:

$$S_{mag} = -k_B n_{Fe} \ln[m(2S+1)] \quad (2)$$

where S is the iron spin quantum number (S = 5/2 for HS and S = 1/2 for LS), and m is the electronic configuration degeneracy (m = 1 for HS and m = 3 for LS).

Placing defects in a crystal also creates configurational entropy. This configurational entropy can be calculated by using Boltzmann's entropy formula:

$$S_{conf} = k_B \ln \Omega \quad (3)$$

$$\Omega = \frac{N!}{(n_i)!(N - n_i)!} \tag{4}$$

where $\Omega$ represents the number of possible configurations of the impurity atom, $N$ is the number of possible crystallographic sites of the impurity atom, and $n_i$ is the number of impurity atoms ($Fe^{3+}$ and H).

## 3. Results and Discussion

### 3.1. Incorporation of Hydrogen and Ferric

It has been previously reported that ferric iron ($Fe^{3+}$) typically occupies the Mg-site (A-site) in Al-bearing $MgSiO_3$, and in Al-free $MgSiO_3$, ferric iron occupies both the Mg and the Si-site (B-site) [17,52]. In addition, water can be incorporated into a cationic vacancy (Mg site or Si site) in a $MgSiO_3$ lattice as a hydrogen defect via a charge-coupled substitution mechanism [53–55]. Therefore, two different hydrous states were considered in this study: a $[Fe^{3+}\text{-H}]_{Si}$ defect that simulates the interaction of water with a $Fe^{3+}$ in a Si octahedral site (B-site), and an $[Fe^{3+}\text{-H}]_{Mg\text{-}Mg}$ defect which simulates the interaction of water with $Fe^{3+}$ in an Mg dodecahedral site (A-site). The stable local arrangement of the $Fe^{3+}$ and the H atom in $MgSiO_3$ lattice is that for the $[Fe^{3+}\text{-H}]_{Si}$ defect, a $Fe^{3+}$ and an H atom occupy a Si-site by replacing the Si atom with a coupled pair, whereby the H atom bonds with an O1 atom in the $SiO_6$ octahedron (Supplementary Materials Figures S1 and S2). For the $[Fe^{3+}\text{-H}]_{Mg\text{-}Mg}$ defect, a $Fe^{3+}$ atom and an H atom occupy two nearest Mg-site by replacing two Mg atoms respectively, and the H atom also bonds with an O1 atom (Supplementary Materials Figure S3 and S4). For all defect structures, we have eliminated from consideration those structures that have unstable phonon spectrums and unfeasibly high enthalpies [42].

We have calculated the energy of $Fe^{3+}$-bearing $MgSiO_3$ with different $Fe^{3+}$ and H defects in our previous study [42] to evaluate the relative stability of $Fe^{3+}$ and H defects in Fe-bearing systems for both Brg and PPv phases. We assumed that the exchange between $[Fe^{3+}\text{-H}]_{Si}$ and $[Fe^{3+}\text{-H}]_{Mg\text{-}Mg}$ defect in the lattice can be represented by the following reaction:

$$\left(Mg_{1-2x}Fe_xH_x\right)SiO_3 + 3xMgO \Longleftrightarrow Mg\left(Si_{1-x}Fe_xH_x\right)O_3 + xMgSiO_3 \tag{5}$$

The left-hand side of the above reaction represents $Fe^{3+}$ and H in the Mg site ($[Fe^{3+}\text{-H}]_{Mg\text{-}Mg}$), and the right-hand side is $Fe^{3+}$ and H in the Si site ($[Fe^{3+}\text{-H}]_{Si}$). The reaction enthalpies $\Delta H$ of reaction (5) have been calculated in our recent study [42]. For both the Brg and PPv phase, the reaction enthalpies decrease with increasing pressure, and the values of $\Delta H$ are negative at all lower mantle pressures, indicating favorably for $[Fe^{3+}\text{-H}]_{Si}$. This was calculated at $x = 0.0625$ but likely holds for all concentrations as $\Delta H$ should be roughly linear with concentration and should not change significantly until very high concentrations are reached, and the entropy is the same if Fe-H is in a Si or an Mg site. Thus, we predict that, in a hydrous $Fe^{3+}$-bearing $MgSiO_3$ system, the $[Fe^{3+}\text{-H}]_{Si}$ defect is energetically more stable compared with $[Fe^{3+}\text{-H}]_{Mg\text{-}Mg}$ defect at all lower mantle pressures and conditions in both Brg and PPv.

### 3.2. Vibrational Properties

The vibrational properties of $[Fe^{3+}\text{-H}]_{Si}$-Brg/PPv and $[Fe^{3+}\text{-H}]_{Mg\text{-}Mg}$-Brg/PPv are investigated by calculating the phonon dispersion curves and vibrational density of states (VDoS) up to 150 GPa, and all results are given in Figures 1 and 2 and Figures S5–S8. The pure Brg and PPv phase has 20 atoms per primitive cell, so there are 60 vibrational modes at each $q$ point in the Brillouin zone. However, the presence of ferric and hydrogen will break the symmetry of lattice and lead to 243 and 240 vibrational modes in the $[Fe^{3+}\text{-H}]_{Si}$ (81 atoms) and $[Fe^{3+}\text{-H}]_{Mg\text{-}Mg}$ (80 atoms) supercells, respectively. In general, the addition of Fe has been predicted to soften the acoustic phonon modes [27], and we predict that this will still occur, especially at the T and R points. For example, at 60 GPa, the lowest acoustic phonon frequency at the T point is softened, from ~191.5 $cm^{-1}$ and ~185.2 $cm^{-1}$

for pure-Brg and pure-PPv to ~116.1 cm$^{-1}$ and ~96.0 cm$^{-1}$ for [Fe$^{3+}$-H]$_{Si}$-Brg and PPv, respectively. In addition, at the R point and at the same pressure, the lowest frequency decreases from ~225.8 cm$^{-1}$ for pure-Brg and ~208.0 cm$^{-1}$ for pure-PPv to ~110.7 cm$^{-1}$ for [Fe$^{3+}$-H]$_{Si}$-Brg and ~96.3 cm$^{-1}$ for [Fe$^{3+}$-H]$_{Si}$-PPv.

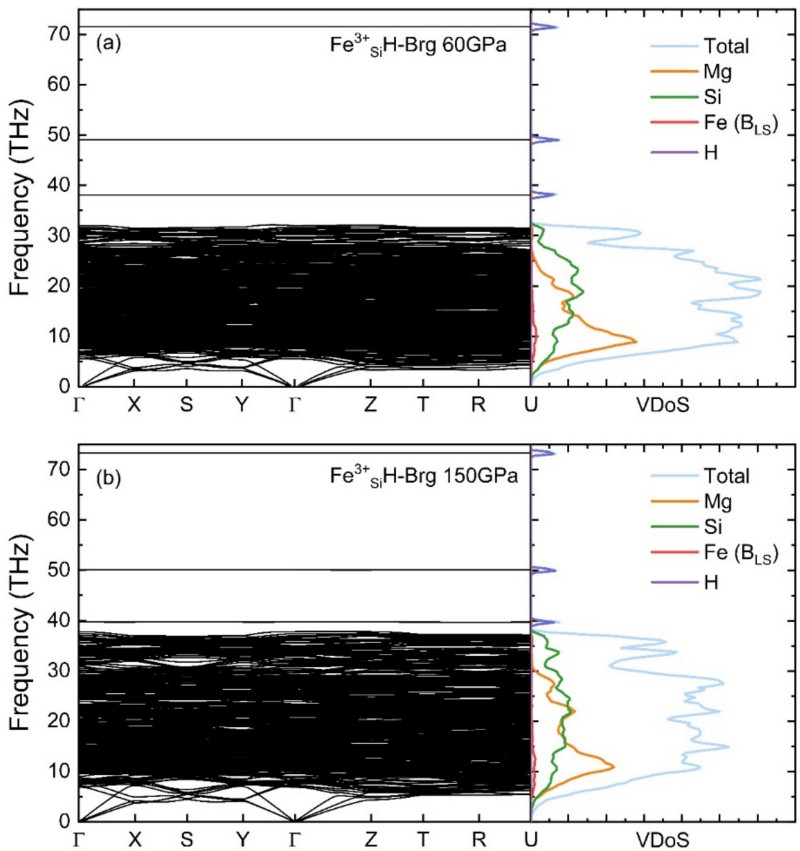

**Figure 1.** Phonon dispersion curves and vibrational density of states (VDoS) of [Fe$^{3+}$-H]$_{Si}$-Brg with LS Fe$^{3+}$ at 60 GPa (**a**) and 150 GPa (**b**), respectively. B$_{LS}$ indicates low spin Fe$^{3+}$ in B site (Si site). Total VDoS and partial VDoS of Mg, Si, Fe (LS), H are shown by light blue, orange, green, red, and purple, respectively.

The phonon dispersion curves and partial VDoS at 60 and 150 GPa for [Fe$^{3+}$-H]$_{Si}$-Brg with LS and HS ferric iron are shown in Figure 1 and Supplementary Materials Figure S5, and for [Fe$^{3+}$-H]$_{Si}$-PPv with LS and HS ferric are shown in Figures S6 and S7, respectively. Both [Fe$^{3+}$-H]$_{Si}$ and [Fe$^{3+}$-H]$_{Mg-Mg}$ bridgmanites contain only real frequencies, which indicates these structures are dynamically stable across lower mantle pressures (>30 GPa). Under 30 GPa, the PPv structures have negative frequencies as PPv is unfavored by low pressure. In [Fe$^{3+}$-H]$_{Si}$-Brg, the phonon dispersion curves and partial VDoS indicate that Fe$^{3+}$ mainly affects the low phonon frequency parts of the spectrum and that three distinctive phonons are created at high frequencies, which uniquely identify the [Fe$^{3+}$-H]$_{Si}$ structure. According to the phonon dispersion curves at different pressures (i.e., Figure 1a,b), the phonon frequencies are increased with increasing pressure at nearly every *q* point, which is consistent with the previous study of dry Fe-bearing bridgmanite [29]. The high-frequency phonons that are related to hydrogen are also affected by the spin state of Fe$^{3+}$, as displayed in Figure 1b and Supplementary Materials Figure S5b. At 150 GPa, the highest frequency optic mode at the Γ point is slightly increased from ~72.2 THz (~2408.4 cm$^{-1}$) for HS to ~73.3 THz (~2445.1 cm$^{-1}$) for LS. In [Fe$^{3+}$-H]$_{Si}$-PPv, Fe$^{3+}$ and hydrogen still contribute to the low and high phonon frequency parts of the system, respectively (Figures S6 and S7). The response of phonon frequencies to pressure is similar to the case of [Fe$^{3+}$-H]$_{Si}$-Brg; with increasing pressure, the phonon frequencies of [Fe$^{3+}$-

H]$_{Si}$-PPv are raised. The effect of the $Fe^{3+}$ spin state on the phonon frequencies of hydrogen in [$Fe^{3+}$-H]$_{Si}$-PPv is opposite to that of [$Fe^{3+}$-H]$_{Si}$-Brg. The highest phonon frequency of hydrogen at the Γ point is decreased from ~64.1 THz (~2138.1 cm$^{-1}$) for an HS ferric iron to ~58.7 THz (~1958.1 cm$^{-1}$) for LS at 150 GPa (Figures S6b and S7b).

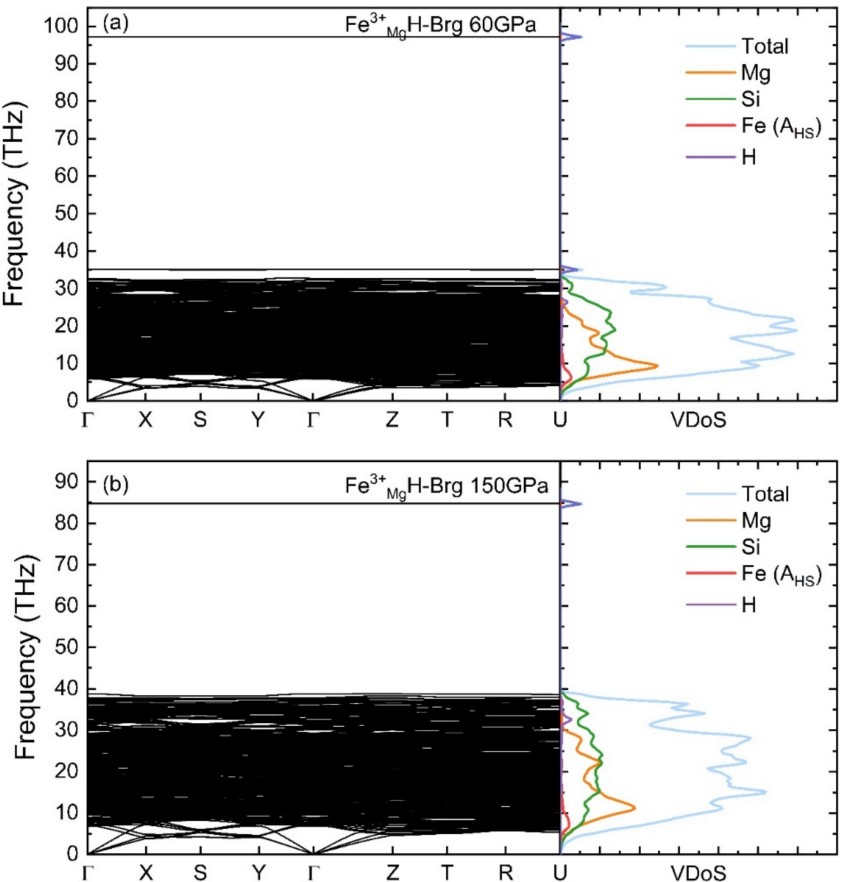

**Figure 2.** Phonon dispersion curves and vibrational density of states (VDoS) of [$Fe^{3+}$-H]$_{Mg-Mg}$-Brg at 60 GPa (**a**) and 150 GPa (**b**), respectively. A$_{HS}$ indicates high spin $Fe^{3+}$ in A site (Mg site). Total VDoS and partial VDoS of Mg, Si, Fe (HS), H are shown by light blue, orange, green, red, and purple, respectively.

In the case of [$Fe^{3+}$-H]$_{Mg-Mg}$ defect configuration, the phonon dispersion curves and partial VDoS at 60 and 150 GPa for [$Fe^{3+}$-H]$_{Mg-Mg}$-Brg and [$Fe^{3+}$-H]$_{Mg-Mg}$-PPv are displayed in Figure 2 and Supplementary Materials Figure S8, respectively. The presence of $Fe^{3+}$ and hydrogen also affect the low and high phonon frequency parts in both [$Fe^{3+}$-H]$_{Mg-Mg}$-Brg and [$Fe^{3+}$-H]$_{Mg-Mg}$-PPv. In [$Fe^{3+}$-H]$_{Mg-Mg}$-Brg, the frequency of the phonons with low and middling phonon frequencies parts of the system increase with increasing pressure but the H-associated phonons with high frequencies decrease with pressure, as shown in Figure 2a,b. The highest hydrogen phonon frequencies at the Γ point are ~97.2 THz (~3242.2 cm$^{-1}$) for 60 GPa and ~84.7 THz (2825.3 cm$^{-1}$) for 150 GPa. These are considerably larger than those seen for the [$Fe^{3+}$-H]$_{Si}$ defect configuration and provide a way of distinguishing between these configurations. In [$Fe^{3+}$-H]$_{Mg-Mg}$-PPv, the frequencies have similar responses to pressure as with [$Fe^{3+}$-H]$_{Mg-Mg}$-Brg, and also have higher phonon frequencies than those of [$Fe^{3+}$-H]$_{Si}$-PPv (Supplementary Materials Figure S8a,b).

### 3.3. Thermodynamic Properties

On the base of calculated phonon frequencies, the Gibbs free energy can be obtained within the QHA frame, then the fundamental thermodynamic properties for $[Fe^{3+}-H]_{Si}$-Brg/PPv and $[Fe^{3+}-H]_{Mg-Mg}$-Brg/PPv are derived from the Helmholtz free energy with standard thermodynamic relations at specific temperatures and pressures [56]. The isothermal bulk modulus $K_T$, thermal expansion coefficient $\alpha$, isochoric $C_V$ and isobaric heat capacities $C_P$, Grüneisen parameter $\gamma$ and the vibrational contribution to the entropy $S_{vib}$ are shown in Figure 3 and Supplementary Materials Table S1 for the Brg-phase, and Figure 4 and Supplementary Materials Table S2 for the PPv-phase, respectively.

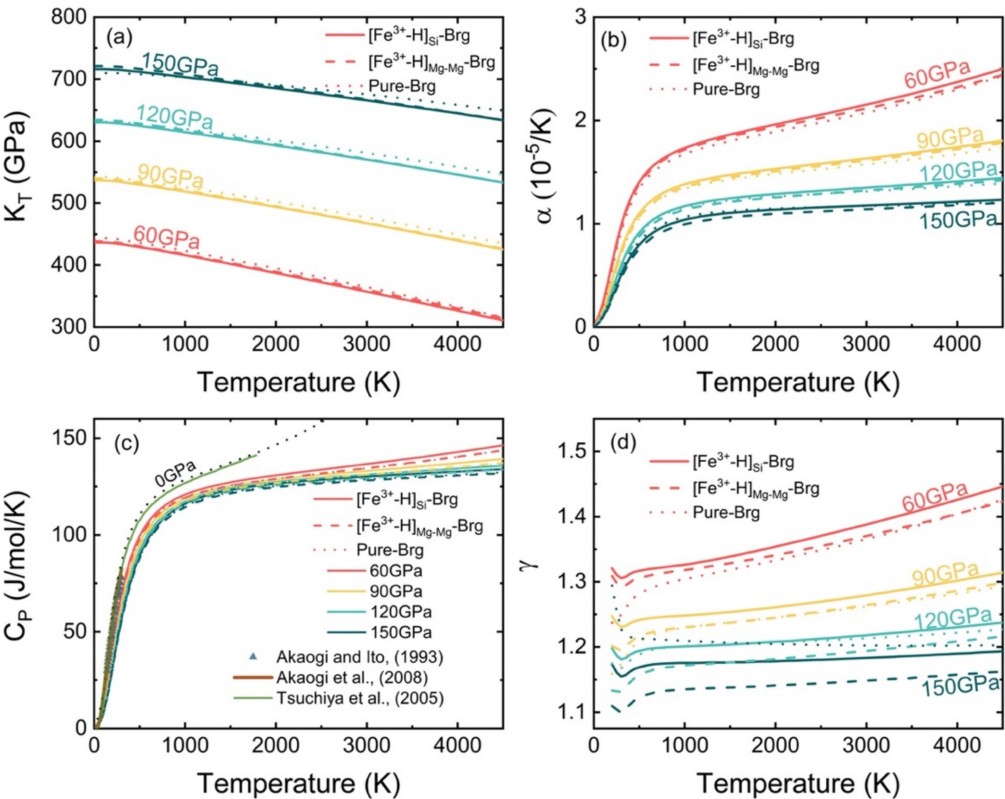

**Figure 3.** The thermodynamic properties of hydrous $Fe^{3+}$-bearing Brg as a function of temperature at different pressures. (**a**) Isothermal bulk modulus, (**b**) Thermal expansivity, (**c**) isobaric heat capacity, (**d**) Grüneisen parameter. Solid, dashed, and dotted lines represent the $[Fe^{3+}-H]_{Si}$-Brg, $[Fe^{3+}-H]_{Mg-Mg}$-Brg, and Pure-Brg, respectively. The experimental values (blue triangle and brown line) are from Akaogi and Ito [57] and Akaogi et al. [58]. The previous theoretical results (green line) are from Tsuchiya et al. [25].

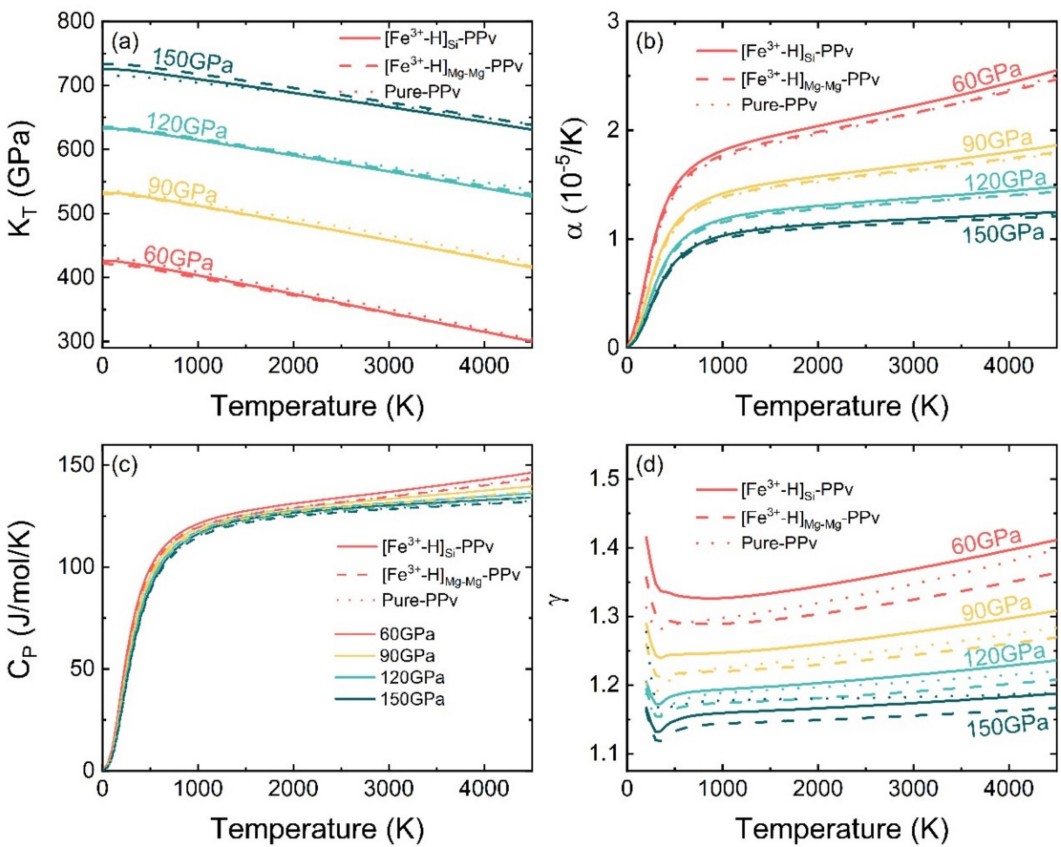

**Figure 4.** The thermodynamic properties of hydrous $Fe^{3+}$-bearing PPv as a function of temperature at different pressures. (**a**) Isothermal bulk modulus, (**b**) Thermal expansivity, (**c**) isobaric heat capacity, (**d**) Grüneisen parameter. Solid, dashed, and dotted lines represent the $[Fe^{3+}-H]_{Si}$-PPv, $[Fe^{3+}-H]_{Mg-Mg}$-PPv, and Pure-PPv, respectively.

At finite temperature, $Fe^{3+}$ can be stabilized as a mixed spin state (MS) of high and low spins, and this can occur at lower mantle conditions. To calculate the thermodynamic properties when the iron is in a mixed spin state, we took the values of the properties calculated at HS and LS and calculated an average weighted by $n_{LS}$ obtained from Equation (1). Calculated low spin state fraction ($n_{LS}$) of $Fe^{3+}$ in $[Fe^{3+}-H]_{Si}$-Brg and $[Fe^{3+}-H]_{Si}$-PPv at specific pressures and temperatures are listed in Table 1.

**Table 1.** Calculated low spin state fraction ($n_{LS}$) of Ferric ($Fe^{3+}$) in $[Fe^{3+}-H]_{Si}$-Brg and $[Fe^{3+}-H]_{Si}$-PPv, respectively, at specific pressure and temperature.

| Configuration | Temperature | 60 GPa | 90 GPa | 120 GPa | 150 GPa |
|---|---|---|---|---|---|
| $[Fe^{3+}-H]_{Si}$-Brg | 300 K | 1 | 1 | 1 | 1 |
| | 1000 K | 0.94 | 1 | 1 | 1 |
| | 2000 K | 0.62 | 0.97 | 1 | 1 |
| | 3000 K | 0.41 | 0.86 | 0.97 | 0.99 |
| | 4000 K | 0.29 | 0.72 | 0.91 | 0.97 |
| $[Fe^{3+}-H]_{Si}$-PPv | 300 K | 1 | 1 | 1 | 1 |
| | 1000 K | 1 | 1 | 1 | 1 |
| | 2000 K | 0.92 | 0.99 | 1 | 1 |
| | 3000 K | 0.72 | 0.94 | 0.99 | 1 |
| | 4000 K | 0.56 | 0.83 | 0.94 | 0.98 |

The isothermal bulk moduli $K_T$ of the $Fe^{3+}$-bearing hydrous system as a function of temperature at a specific pressure are displayed in Figure 3a for Brg-phase and Figure 4a for PPv-phase. For the Brg-phase (Figure 3a and Supplementary Materials Table S1), the presence of ferric and hydrogen reduce the $K_T$ compared with pure-Brg at the lower mantle

P-T range. For instance, at 90 GPa and 2000 K, $[Fe^{3+}-H]_{Si}$-Brg and $[Fe^{3+}-H]_{Mg-Mg}$-Brg reduce the $K_T$ by ~1.7% and ~1.5% compared to pure-Brg, respectively. Meanwhile, this contrast on the $K_T$ will increase with temperature for both $[Fe^{3+}-H]_{Si}$-Brg and $[Fe^{3+}-H]_{Mg-Mg}$-Brg. At 4000 K, the differences of $K_T$ between $[Fe^{3+}-H]$-bearing and pure system are raised to −2.2% for $[Fe^{3+}-H]_{Si}$-Brg and −1.9% for $[Fe^{3+}-H]_{Mg-Mg}$-Brg. On the other hand, the contrast of $K_T$ between $[Fe^{3+}-H]_{Si}$-Brg and $[Fe^{3+}-H]_{Mg-Mg}$-Brg is marginal, which indicates that the $K_T$ is only slightly affected by $[Fe^{3+}-H]$ defect configuration ($[Fe^{3+}-H]$ at Mg site or Si site). This data can be compared to that of pure iron, which is ~2.0% [29] for the same concentration of iron. The effect of hydrogen is, therefore, to lessen the effect of iron on the modulus and allows iron to be partitioned into bridgmanite without having a large effect on the seismic properties.

For PPv, its isothermal bulk moduli $K_T$ is also decreased by the incorporation of $Fe^{3+}$ and hydrogen (Figure 4a and Supplementary Materials Table S2). Compared to the pure-PPv, at 120 GPa and 2000 K, $[Fe^{3+}-H]_{Si}$ and $[Fe^{3+}-H]_{Mg-Mg}$ defect reduce the $K_T$ of the PPv-phase by ~0.7% and ~0.3%, respectively. This effect reverses with pressure; however, at 150 GPa, $[Fe^{3+}-H]_{Si}$-PPv has a $K_T$ that is ~0.8% higher than that of pure-PPv when the temperature is lower than 2000 K. The increasing temperature lowers the $K_T$ of the hydrous PPv, which is the same with Brg. Similarly, the $K_T$ of $[Fe^{3+}-H]_{Mg-Mg}$-PPv is larger than pure-PPv's at 150 GPa and the temperature of up to 3900 K.

In general, Fe-H only makes a small change to the $K_T$ of Brg and PPv and a smaller change than that to pure iron, softening its effect. In the deep mantle where Fe-H is expected to be relevant, the presence of such defects reduces the $K_T$ of both Brg and PPv due to the high temperature. The $K_T$ of Brg is reduced more than the $K_T$ of PPv by ~0.8%. If we scale this to an expected mantle composition of ~1000 wt. ppm then this induces an anomaly of 0.2% across the Brg to PPv phase transition.

The thermal expansion coefficient $\alpha$ of the Brg-phase and PPv-phase at different pressures are determined by $\alpha = 1/V(\partial V/\partial T)_P$, and are plotted in Figures 3b and 4b as a function of temperature. The detailed values of the thermal expansion coefficient are listed in Supplementary Materials Tables S1 and S2. At lower mantle conditions such as at 90 GPa and 2000 K, the $\alpha$ of $[Fe^{3+}-H]_{Si}$-Brg and $[Fe^{3+}-H]_{Mg-Mg}$-Brg are ~$1.53 \times 10^{-5}$ $K^{-1}$ and ~$1.51 \times 10^{-5}$ $K^{-1}$, respectively, which are larger than that of pure-Brg with a value of $1.48 \times 10^{-5}$ $K^{-1}$ (Supplementary Materials Table S1). With increasing pressure, the difference between the hydrous and the pure system is reduced (Figure 3b), at 150 GPa and 2000 K, $[Fe^{3+}-H]_{Si}$-Brg has an $\alpha$ which almost equal to the value of pure-Brg. In addition, compared to the $[Fe^{3+}-H]_{Mg-Mg}$-Brg, $[Fe^{3+}-H]_{Si}$-Brg always has a larger $\alpha$ at the investigated P-T range. This has a small effect of around 1.0% on the density of the system. This is important because a small change of density is needed for LLSVPs. Large changes in density will call large changes to seismic velocities and likely get segregated to the bottom of the mantle. Very similar trends are seen in the PPv phase with $[Fe^{3+}-H]_{Si}$-PPv increasing $\alpha$ from ~$1.28 \times 10^{-5}$ $K^{-1}$ to ~$1.31 \times 10^{-5}$ $K^{-1}$ at 120 GPa and 2000 K but this increase decreases with pressure while $[Fe^{3+}-H]_{Mg-Mg}$ makes no large change to $\alpha$. Across the Brg to PPv transition, the presence of $[Fe^{3+}-H]_{Si}$ makes small difference to the density change of the transition (~1.3% at D'' layer depth) and thus does not substantially impact the rheological stability of the D'' layer.

The isobaric heat capacities $C_P$ calculated from $C_P = -T(\partial^2 G/\partial T^2)_P$ are shown in Figure 3c and Supplementary Materials Table S1 for the Brg-phase and Figure 4c and Supplementary Materials Table S2 for the PPv-phase, respectively. The isobaric heat capacities of pure-Brg calculated at 0 GPa are in good agreement with the results from experimental studies [57,58], especially at around 300 K. Moreover, comparing the $C_P$ we calculated and the data from other theoretical study, the value of $C_P$ of pure-Brg at 0 GPa are fairly close to the results predicted by Tsuchiya, Tsuchiya, and Wentzcovitch [25]. This indicate that the thermodynamic parameters we predicted in this study are reasonable. At the pressure higher than 0 GPa, the $C_P$ of the Brg-phase is increased by the incorporation of $Fe^{3+}$ and hydrogen via a $[Fe^{3+}-H]_{Si}$ defect, especially at temperatures higher than 2500 K.

However, the Brg-phase's $C_P$ is not modified by the presence of the $[Fe^{3+}-H]_{Mg-Mg}$ defect (change is on average $\sim -0.1\%$) (Figure 3c and Supplementary Materials Table S1). In the case of the PPv-phase, $[Fe^{3+}-H]_{Si}$-PPv has a larger $C_P$ (127.5 J·mol$^{-1}$·K$^{-1}$) than pure-PPv (125.9 J·mol$^{-1}$·K$^{-1}$) at 120 GPa and 2000 K, and the $C_P$ of $[Fe^{3+}-H]_{Mg-Mg}$-PPv is almost near that of pure-PPv at lower mantle conditions (Figure 4c and Supplementary Materials Table S2). This tendency is the same as those of the Brg-phase. The isochoric heat capacity $C_V$ in all systems decreases with pressure when the temperature is lower than 2000 K, but at the mantle, relevant temperatures $C_V$ become insensitive to pressure changing (Supplementary Materials Figures S9 and S10). Moreover, the $[Fe^{3+}-H]_{Si}$ defect can yield a large isochoric heat capacity than the $[Fe^{3+}-H]_{Mg-Mg}$ defect and pure system whether in the Brg- or PPv-phase (Supplementary Materials Figures S9 and S10). The thermodynamic Grüneisen parameter $\gamma$ ($\gamma = \alpha V K_T / C_V$) of the Brg-phase and the PPv-phase are displayed in Figures 3d and 4d and Tables S1 and S2, respectively. The $\gamma$ of $[Fe^{3+}-H]_{Si}$-Brg is larger than that of $[Fe^{3+}-H]_{Mg-Mg}$-Brg at all investigated pressures and temperatures, as shown in Figure 3d and Supplementary Materials Table S1. At pressures lower than 90 GPa, pure-Brg has a smaller or almost equal $\gamma$ compared to the hydrous $Fe^{3+}$-bearing systems; however, when pressure is higher than 90 GPa, the $\gamma$ of pure-Brg becomes significantly higher than that of both $[Fe^{3+}-H]_{Si}$-Brg and $[Fe^{3+}-H]_{Mg-Mg}$-Brg, particularly at 150 GPa. The $\gamma$ of all systems is gradually increased when the temperature increases, and this effect of temperature is increasingly suppressed at high pressures. As for the PPv-phase, compared to pure-PPv, the incorporation of hydrogen and $Fe^{3+}$ via the $[Fe^{3+}-H]_{Si}$ defect increases the $\gamma$ of the PPv-phase significantly at pressures below 150 GPa, but these two values converge around 150 GPa and 4000 K. Unlike $[Fe^{3+}-H]_{Si}$-PPv, the presence of an $[Fe^{3+}-H]_{Mg-Mg}$ defect always decreases the $\gamma$ of the PPv-phase at all pressures and temperatures that we studied (Figure 4d and Supplementary Materials Table S2). Furthermore, the $\gamma$ of $[Fe^{3+}-H]_{Mg-Mg}$-PPv is also less than $[Fe^{3+}-H]_{Si}$-PPv observable across the lower mantle range. We find that the effects of H and $Fe^{3+}$ defect on the fundamental thermodynamic properties (such as $K_T$, $\alpha$, $C_P$) of $MgSiO_3$ are moderate due to a low H and $Fe^{3+}$ concentration in structure, but that the Grüneisen parameter $\gamma$ is sensitive to the presence of hydrogen and ferric atoms. The $Fe^{3+}$-H defect will yield the change with up to $\sim 6.5\%$ on the Grüneisen parameter. With a typical mantle concentration of 1000 wt. ppm this change is $\sim 1.2\%$. This may imply that even a small amount of water may affect the anharmonicity of $Fe^{3+}$-bearing $MgSiO_3$ in lower mantle conditions. This will also affect the adiabaticity of the mantle in the presence of sufficient Fe and H. The anharmonicity of the $MgSiO_3$ system when water is presented needs further consideration.

The vibrational contribution to the entropy $S_{vib}$ as a function of temperature at different pressures is displayed in Supplementary Materials Figure S11 and Table S1 for the Brg-phase and Supplementary Materials Figure S12 and Table S2 for the PPv-phase. In Figures S11 and S12, we see that vibrational entropy of the system is increased with the temperature increase but decreased with pressure. This variation trend is consistent with the results of a previous study on the dry $MgSiO_3$ or transition zone minerals [27,28,59–61]. In both Brg and PPv-phase, $[Fe^{3+}-H]_{Si}$ defect will produce a high vibrational entropy, whereas the vibrational entropy of the system is less affected by the incorporation of $[Fe^{3+}-H]_{Mg-Mg}$, further showing that $[Fe^{3+}-H]_{Si}$ is the stable defect in these systems. The difference between $[Fe^{3+}-H]_{Si}$ containing Brg and PPv is $\sim 2.1$ J mol$^{-1}$ K$^{-1}$ at conditions close to the D″ (127 GPa and 2500 K) and thus in the presence of Fe and H Brg is further favored by temperature than in PPv, which will serve to enhance the possibility of double-crossing though a full phase diagram needs to be constructed at this point.

Our entropy calculation indicates that, whether in bridgmanite or post-perovskite, the $[Fe^{3+}-H]_{Si}$ configuration will produce a higher entropy than the $[Fe^{3+}-H]_{Mg-Mg}$ configuration at the P-T conditions of the lower mantle. This means that hydrous and ferric iron will more preferentially be substituted into the Brg and PPv via a $[Fe^{3+}-H]_{Si}$ defect. Our previous study found that the $[Fe^{3+}-H]$ defect would shift the PPv phase transition boundary [42]. We can conclude in this study that, when hydrogen and ferric iron are

presented in the deep lower mantle, the phase transition boundary between Brg and PPv will prefer to move to the shallower part of the lower mantle, and the ridge region of the D″ layer might be rich in Hydrogen and iron.

Temperature distribution in the deep mantle is important for constraining the thermal structure of the lower mantle. Estimation of the temperature profile in the lower mantle can help us determine the temperature distribution in this region and further better understand the thermal structure of the lower mantle. It is generally accepted that the adiabatic temperature profile can represent the variation of temperature in the Earth's mantle, and the adiabatic temperature gradient is expressed as the equation $(dT/dP)_S = T\gamma/K_S$. By assuming the lower mantle is approximated as the pyrolite composition with a mixture of ~80% bridgmanite with $[Fe^{3+}-H]_{Si}$ and ~20% (Mg, Fe)O, we estimated the average adiabatic temperature gradient in the lower mantle is ~0.43 K/km, which is higher than the results determined in the dry lower mantle of ~0.27–~0.32 K/km [57,62,63]. We obtained the temperature at the pressure of ~127 GPa, which corresponding to the top of the D″ layer is ~2640 K, which is higher than the temperature of ~2500 K proposed by the previous studies [57,64]. The presence of water will make the adiabatic temperature profile in the lower mantle steeper than that of the dry condition. Moreover, at the top of the D″ layer, those regions where the dominant component is hydrous $Fe^{3+}$-bearing bridgmanite might have a higher temperature than the ambient lower mantle.

## 4. Conclusions

In this paper, we have investigated the vibrational and thermodynamic properties of hydrous iron-bearing bridgmanite and post-perovskite with different $[Fe^{3+}-H]$ configurations. The phonon dispersion curves of different hydrous configurations were calculated using GGA+U and the density functional perturbation theory (DFPT) method. The thermodynamic properties were determined using quasi-harmonic approximation (QHA) calculations.

In hydrous iron-bearing $MgSiO_3$ systems, ferric ($Fe^{3+}$) iron mainly affects the low phonon frequency parts of the system; however, the incorporation of hydrogen contributes to the highest phonon frequency parts. Water and iron incorporation will lead to vibrational instability of $MgSiO_3$ at pressures lower than 30 GPa. Furthermore, the variation of phonon frequencies with pressure is strongly affected by the different site occupancy of H and $Fe^{3+}$ in the lattice (i.e., in the Si site or Mg site).

The presence of ferric and hydrogen can affect the thermodynamic properties of the $MgSiO_3$ system further. Of all the defect configurations that we considered, $[Fe^{3+}-H]_{Si}$ defects produce the largest effect on the thermodynamic properties of Brg and PPv compared to $[Fe^{3+}-H]_{Mg-Mg}$ defects and the pure system. $[Fe^{3+}-H]_{Si}$ defects increase the fundamental thermodynamic parameters, including isothermal bulk modulus $K_T$, thermal expansion coefficient $\alpha$, heat capacity $C_P$, $C_V$ and vibrational contribution to the entropy $S_{vib}$ for both Brg and PPv. The $[Fe^{3+}-H]_{Mg-Mg}$ defect, however, only makes slight changes to the above thermodynamic parameters. In addition, among the thermodynamic parameters we studied, the Grüneisen parameter $\gamma$ is the most sensitive to the incorporation of hydrogen and ferric iron but the variation trends of $\gamma$ with defect configuration, pressure and temperature is complex. Although we have studied the effects of water, defect configurations, temperature, and pressure on the vibrational and thermodynamic properties of bridgmanite and post-perovskite, the influence of concentration of water and iron still needs to be further studied.

**Supplementary Materials:** The following are available online at https://www.mdpi.com/article/10.3390/min11080885/s1, Figure S1: Crystal structures in details of $[Fe^{3+}-H]_{Si}$-Brg. In lattice structure, orange, dark blue, red, grey, and black spheres represent magnesium, silicon, oxygen, iron, and hydrogen, respectively. Figure S2: Crystal structures in details of $[Fe^{3+}-H]_{Si}$-PPv. In lattice structure, orange, dark blue, red, grey, and black spheres represent magnesium, silicon, oxygen, iron, and hydrogen, respectively. Figure S3: Crystal structures in details of $[Fe^{3+}-H]_{Mg-Mg}$-Brg. In lattice structure, orange, dark blue, red, grey, and black spheres represent magnesium, silicon, oxygen, iron,

and hydrogen, respectively. Figure S4: Crystal structures in details of $[Fe^{3+}-H]_{Mg-Mg}$-PPv. In lattice structure, orange, dark blue, red, grey, and black spheres represent magnesium, silicon, oxygen, iron, and hydrogen, respectively. Figure S5: Phonon dispersion curves and vibrational density of states (VDoS) of $[Fe^{3+}-H]_{Si}$-Brg with HS $Fe^{3+}$ at 60 GPa (a) and 150 GPa (b). $B_{HS}$ indicates that the high spin $Fe^{3+}$ in the B site (Si site). Total VDoS and partial VDoS of Mg, Si, Fe (HS), H are shown by light blue, orange, green, red, and purple, respectively. Figure S6: Phonon dispersion curves and vibrational density of states (VDoS) of $[Fe^{3+}-H]_{Si}$-PPv with LS $Fe^{3+}$ at 60 GPa (a) and 150 GPa (b). $B_{LS}$ indicates that the low spin $Fe^{3+}$ in B site (Si site). Total VDoS and partial VDoS of Mg, Si, Fe (LS), H are shown by light blue, orange, green, red, and purple, respectively. Figure S7: Phonon dispersion curves and vibrational density of states (VDoS) of $[Fe^{3+}-H]_{Si}$-PPv with HS $Fe^{3+}$ at 60 GPa (a) and 150 GPa (b). $B_{HS}$ indicates that the high spin $Fe^{3+}$ in the B site (Si site). Total VDoS and partial VDoS of Mg, Si, Fe (HS), H are shown by light blue, orange, green, red, and purple, respectively. Figure S8: Phonon dispersion curves and vibrational density of states (VDoS) of $[Fe^{3+}-H]_{Mg-Mg}$-PPv at 60 GPa (a) and 150 GPa (b). $A_{HS}$ indicates that the high spin $Fe^{3+}$ in A site (Mg site). Total VDoS and partial VDoS of Mg, Si, Fe (HS), H are shown by light blue, orange, green, red, and purple, respectively. Figure S9: The heat capacity at constant volume as a function of temperature at different pressures. Solid, dashed, and dotted lines represent the $[Fe^{3+}-H]_{Si}$-Brg, $[Fe^{3+}-H]_{Mg-Mg}$-Brg, and Pure-Brg, respectively. Figure S10: The heat capacity at constant volume as a function of temperature at different pressures. Solid, dashed, and dotted lines represent the $[Fe^{3+}-H]_{Si}$-PPv, $[Fe^{3+}-H]_{Mg-Mg}$-PPv, and Pure-PPv, respectively. Figure S11: Temperature dependence of vibrational entropy at different pressures. Solid, dashed, and dotted lines represent the $[Fe^{3+}-H]_{Si}$-Brg, $[Fe^{3+}-H]_{Mg-Mg}$-Brg, and Pure-Brg, respectively. Figure S12: Temperature dependence of vibrational entropy at different pressures. Solid, dashed, and dotted lines represent the $[Fe^{3+}-H]_{Si}$-PPv, $[Fe^{3+}-H]_{Mg-Mg}$-PPv, and Pure-PPv, respectively. Table S1: Calculated thermodynamic parameters of hydrous iron-bearing and pure structure for bridgmanite. Table S2: Calculated thermodynamic parameters of hydrous iron-bearing and pure structure for post-perovskite.

**Author Contributions:** Conceptualization, F.Z.; methodology, J.J.; formal analysis, J.J. and J.M.R.M.; investigation, J.J.; data curation, J.J.; writing—original draft preparation, J.J.; writing—review and editing, F.Z. and J.M.R.M.; supervision, F.Z.; project administration, F.Z.; funding acquisition, F.Z. All authors have read and agreed to the published version of the manuscript.

**Funding:** This research was funded by the National Natural Science Foundation of China (grant number: 41773057 (01/2018–12/2021), 42050410319 (01/2021–12/2022)).

**Data Availability Statement:** The data that support the findings of this study are available from the corresponding author upon reasonable request.

**Acknowledgments:** The authors thank the computations support of the National Supercomputer Center in Shenzhen, China, and Supercomputer Center in Lvliang, China.

**Conflicts of Interest:** The authors declare no conflict of interest.

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
