# Peer review of "Vibrational and Thermodynamic Properties of Hydrous Iron-Bearing Lowermost Mantle Minerals"

_minerals, doi:10.3390/min11080885_

Round 1

Reviewer 1 Report

The present study brings new theoretical data on the effect of [Fe3+ + H] substitution on the vibrational and thermodynamic properties of the two major lower mantle minerals : MgSiO3 bridgmanite (Brg) and post-perovskite (ppv). These properties are critical to understand the Brg/ppv phase transition as well as to model the thermal structure of deep mantle structure.

Previous studies have focused on the substitution mechanisms of iron and hydrogen in Brg and ppv and their effect on these high pressure minerals properties separately. Here the authors consider two conditions where [Fe3+-H] substitute itself in the Si site (also named B site) or in the Mg site (A site). They report the effect of both the incorporation of ferric iron and hydrogen in the crystal structure as well as the effect of the configuration of these substitution (in which crystallographic site Fe3+-H in incorporated).

Overall this is an interesting study that brings new important datas for the deep mantle community. I find however, that it lacks of discussion/conclusion: it lacks of comparison with previous studies (experimental or theoretical) and the conclusion in terms of phase transition and thermal structures of the deep Earth.

Some small suggestions :

line 161 : it should read “ferric iron”

line 171 no dot '.’ after Si06

line 201 it will be great to provide values of Fe-free Brg and ppv in order to realize how much acoustic phonon are softer.

Supplementary Fig S1-S4 : it would be easier to visualize the crystallographic structure changes if the unit cell was presented completely

Supplementary Fig S5-S8 legends : “respectively” is not need in the first sentence

Author Response

Response to Reviewer 1 Comments

We thank this reviewer's positive comments and the constructive suggestions for improving our manuscript.

Point 1: I find however, that it lacks of discussion/conclusion: it lacks of comparison with previous studies (experimental or theoretical) and the conclusion in terms of phase transition and thermal structures of the deep Earth.

Response 1: The comparison with previous experimental or theoretical studies has been added in the revision (Line 333-338)

Geophysical implication for phase transition and thermal structures of the deep Earth has been discussed (Line 385-409).

Point 2: line 161 : it should read “ferric iron”.

Response 2: This has been done.

Point 3: line 171 no dot '.’ after Si06.

Response 3: The dot ‘.’ after Si06 has been removed.

Point 4: line 201 it will be great to provide values of Fe-free Brg and ppv in order to realize how much acoustic phonon are softer.

Response 4: This has been done (Line 202-207)

We have provided the lowest acoustic phonon frequency of pure-Brg, pure-PPv, [Fe3+-H]Si-Brg and [Fe3+-H]Si-PPv for both T and R points at 60 GPa

Point 5: Supplementary Fig S1-S4 : it would be easier to visualize the crystallographic structure changes if the unit cell was presented completely.

Response 5: This has been done in Fig S1-S4. The complete unit cell structures are provided in the revision.

Point 6: Supplementary Fig S5-S8 legends : “respectively” is not need in the first sentence.

Response 6: It has been removed.

Reviewer 2 Report

This paper investigates the vibrational and thermodynamics properties of bridgmanite and post-perovskite with hydrous-Fe impurities.

The paper is technically sound, calculations are well executed, results are carefully analyzed and presented, the manuscript is well written, and the conclusions very relevant for advancing understanding the lower mantle. It will be a good addition to the lower mantle literature.

I don’t see any major breakthrough in understanding the properties of the lower mantle, but nevertheless, this paper advances the understanding of lower mantle phases. Perhaps it would be good to move Figs. S1-S4 to the main text. I am sure readers will be interested in seeing the structure of the defects investigated. This is an optional request.

Reviewer 3 Report

The manuscript by Jiang et al. presents vibrational and thermodynamic properties of hydrous Fe3+-bearing bridgmanite and post-perovskite predicted by using first-principles calculations. This study is following the paper, Theoretical studies on the hydrous lower mantle and D” layer minerals, published by the same authors in Earth and Planetary Science Letters, 2019. Since the authors have investigated and published the incorporation mechanisms of H, elastic properties, and phase relations of hydrous Fe3+-bearing bridgmanite and post-perovskite, it is not surprising that the vibrational properties of these phases can be calculated and a series of thermodynamic parameters can be derived. Generally speaking, this study adds some new thermodynamic parameters of hydrous lower mantle silicates into the database, which might be used by mineralogists in further studies of Earth’s lower mantle. I support it for publication in Minerals after addressing the following concerns:

  1. I confess that I am not qualified to judge the specifics of the DFT calculations, so before publication, other referees with greater knowledge of DFT calculations (including those involving Fe with different spin states, which is generally quite challenging), should be consulted.
  2. The authors only provided the calculated results at 60, 90, 120, 150 GPa, and didn't compare with any experimental measurements. In order to solidify the quality of theoretical predictions, I suggest the authors provide the calculated results at 0 GPa for both MgSiO3 bridgmanite and H-Fe3+-bearing bridgmanite, which can be directly compared to experimental results, such as heat capacity:

Akaogi, M. & Ito, E. Heat-Capacity of MgSiO3 Perovskite. Geophys Res Lett 20, 105-108, (1993).

Akaogi, M., Kojitani, H., Morita, T., Kawaji, H. & Atake, T. Low-temperature heat capacities, entropies and high-pressure phase relations of MgSiO3 ilmenite and perovskite. Phys Chem Miner 35, 287-297 (2008).

  1. In the introduction, the authors mentioned a lot about D” layer region, bridgmanite phase transition, water in the lower mantle, etc, as motivations of this study. For example, in lines 83-86, the author mentioned, “influence of water on the vibrational and thermodynamic properties of bridgmanite and post-perovskite need to be determined further in order to better understand the thermodynamic and thermal structure of the lower mantle”. However, none of these questions is quantitatively addressed at the end of the paper, and the readers may be curious about how to better understand the lower mantle based on the present study. I suggest the author either add a geophysical implication section using the obtained parameters, or reorganize the introduction part from the perspectives of thermodynamics and materials.

Author Response

Response to Reviewer 2 Comments

We thank this reviewer for their comments and the recommendation of publication on Minerals.

Point 1: I confess that I am not qualified to judge the specifics of the DFT calculations, so before publication, other referees with greater knowledge of DFT calculations (including those involving Fe with different spin states, which is generally quite challenging), should be consulted.

Response 1: The quality of our DFT calculations can be justified by comparing our data with the previous results reported by other DFT experts. For example, the thermodynamic parameters of pure system, spin state fraction and phonon dispersion curves are all in well agreement with the previous results predicted by DFT calculation, such as Tsuchiya and Wang et al, 2013; Tsuchiya and Wentzcovitch et al., 2006; Metsue and Tsuchiya, 2011&2012. Hsu et al., 2011&2012. Therefore, we have confidence to use the same methodology to calculate the same system with Fe and Hydrogen as defect.

Point 2: I suggest the authors provide the calculated results at 0 GPa for both MgSiO3 bridgmanite and H-Fe3+-bearing bridgmanite, which can be directly compared to experimental results, such as heat capacity:

Akaogi, M. & Ito, E. Heat-Capacity of MgSiO3 Perovskite. Geophys Res Lett 20, 105-108, (1993).

Akaogi, M., Kojitani, H., Morita, T., Kawaji, H. & Atake, T. Low-temperature heat capacities, entropies and high-pressure phase relations of MgSiO3 ilmenite and perovskite. Phys Chem Miner 35, 287-297 (2008).

Response 2: We did calculate the thermodynamic results at 0 GPa in this study, however, there have significant negative phonon frequencies at 0 GPa for both [Fe3+-H]Si-Brg and [Fe3+-H]Si-PPv structure. Therefore, we can’t obtain the exact thermodynamic results of [Fe3+-H]Si-Brg and [Fe3+-H]Si-PPv due to instable structures at 0 GPa.

In the revision, we have updated the Figure 3, and provided the thermodynamic results (heat capacity) of pure-Brg at 0 GPa in the Figure 3c, and we have added the data points from experimental (Akaogi & Ito, 1993; Akaogi et al., 2008) and theoretical (Tsuchiya et al., 2005) studies in Figure 3c for comparison. (Line 333-338).

Point 3: In the introduction, the authors mentioned a lot about D” layer region, bridgmanite phase transition, water in the lower mantle, etc, as motivations of this study. For example, in lines 83-86, the author mentioned, “influence of water on the vibrational and thermodynamic properties of bridgmanite and post-perovskite need to be determined further in order to better understand the thermodynamic and thermal structure of the lower mantle”. However, none of these questions is quantitatively addressed at the end of the paper, and the readers may be curious about how to better understand the lower mantle based on the present study. I suggest the author either add a geophysical implication section using the obtained parameters, or reorganize the introduction part from the perspectives of thermodynamics and materials.

Response 3: Geophysical implication has been quantitatively addressed at the end of the paper, where we discussed the implications for phase transition and thermal structures of the deep Earth (Line 385-409).